# Preparation and Structure of Zinc–Calcium Hydroxyapatite Solid Solution Particles and Their Ultraviolet Absorptive Ability

Akemi Yasukawa *☉ and Minami Yamada

School of Home Economics, Faculty of Education, Hirosaki University, 1-Bunkyo, Hirosaki 036-8560, Japan
* Correspondence: yasukawa@hirosaki-u.ac.jp

**Abstract:** The calcium ions ($Ca^{2+}$) of calcium hydroxyapatite (CaHap) were substituted with zinc ions ($Zn^{2+}$), and zinc–calcium hydroxyapatite solid solution (ZnCaHap) particles were prepared via a precipitation method. The structure of the various obtained particles was investigated via powder X-ray diffraction, field emission scanning electron microscopy and energy dispersive X-ray spectrometry. The ultraviolet (UV) absorption ability of the particles was also investigated using UV–Vis spectroscopy. The morphology of the CaHap comprised fine ellipsoidal particles, and long rod-like particles and large plate-like particles were mixed with the fine particles at higher $Zn^{2+}$ contents in the particles. Pure ZnCaHap particles were obtained from the starting solution at less than $Zn/(Zn + Ca)$ ($[X_{Zn}]$) of 0.25. Another crystal phase was mixed with the ZnCaHap phase at $[X_{Zn}] \geq 0.25$. The crystallinity and lattice parameters $a$ and $c$ of the particles decreased with an increase in $[X_{Zn}]$ from 0 to 0.10. The UV absorptive ability of the particles first increased and then decreased with increasing $Zn^{2+}$ content and showed a maximum at $[X_{Zn}] = 0.30$.

**Keywords:** hydroxyapatite; calcium; zinc; ultraviolet absorption; microscopy

## 1. Introduction

Calcium hydroxyapatite, $Ca_{10}(PO_4)_6(OH)_2$ (CaHap), is a major inorganic component of animal bones and teeth. The ion exchange ability of CaHap is so high that many types of cations, including $Mg^{2+}$, $Sr^{2+}$, $Ba^{2+}$, $Cd^{2+}$, $Pb^{2+}$, $Ti^{4+}$ and some rare earth metals, can be incorporated into CaHap crystals, and various metal–calcium hydroxyapatite (MCaHap) solid solutions can be formed [1–21]. $Sr^{2+}$ and $Cd^{2+}$ ions can be incorporated into CaHap crystals at $M/(M + Ca) = 0–1$, and MCaHap with $M/(M + Ca) = 0–1$ can be formed [3–6]. There are different reports of the formation of barium–calcium hydroxyapatite (BaCaHap). $Ba^{2+}$ ions can be incorporated into CaHap crystals, and BaCaHap particles can be formed at $Ba/(Ba + Ca) = 0–1$ [7], at $Ba/(Ba + Ca) = 0.65–1$ [8] and at $Ba/(Ba + Ca) \leq 0.70$ [9]. It was reported that $Ba_3(PO_4)_2$ particles were mixed with BaCaHap at $0.70 < Ba/(Ba + Ca) < 1$ [9]. There were various reports of the formation of magnesium–calcium hydroxyapatite (MgCaHap). It was reported that the incorporation of Mg into synthetic apatites was very limited (a maximum of about 0.4 wt% Mg) [10]. $Mg^{2+}$ ions can be incorporated into calcium chlorapatite crystals at $Mg/(Mg + Ca) \leq 0.30$ [11]; $Mg^{2+}$ ions can also be incorporated into CaHap crystals; MgCaHap particles can be formed at $Mg/(Mg + Ca) \leq 0.50$ and the products are amorphous at $Mg/(Mg + Ca) > 0.50$ [12]. There were also various reports of the formation of lead–calcium hydroxyapatite (PbCaHap). It was reported that an attempt was made to prepare PbCaHap particles with $Pb/(Pb + Ca) = 0–1$, and a change in the break point of the lattice parameter was shown [13]. $Pb^{2+}$ ions can be incorporated into CaHap crystals, and PbCaHap particles can be formed at $Pb/(Pb + Ca) \leq 0.50$ [14]. In another report, $Pb^{2+}$ ions could be incorporated into CaHap crystals, and PbCaHap particles could be formed at $Pb/(Pb + Ca) \leq 0.60$ and $Pb/(Pb + Ca) \geq 0.90$. A mixture of PbHap, β-tricalcium phosphate (β-TCP) and an unknown crystal phase was formed at $0.60 < Pb/(Pb + Ca) < 0.90$ [15]. $Y^{3+}$ ions can be incorporated into CaHap crystals, and

yttrium–calcium hydroxyapatite (YCaHap) can be formed at $Y/(Y + Ca) \leq 0.10$; a mixture of YCaHap and $YPO_3$ was formed at $Y/(Y + Ca) > 0.10$ [16]. $Ti^{4+}$ ions can be incorporated into CaHap crystals, while titanium–calcium hydroxyapatite (TiCaHap) particles can be formed at $Ti/(Ti + Ca) \leq 0.10$ and amorphous particles can form at $Ti/(Ti + Ca) > 0.10$ [17]. $Pr^{3+}$, $Nd^{3+}$ and $Er^{3+}$ ions can be incorporated into CaHap crystals; rare earth metal–calcium hydroxyapatite (LnCaHap) particles can be formed at $Ln/(Ln + Ca) \leq 0.03$ and $LnPO_4$ was mixed with LnCaHap at $Ln/(Ln + Ca) > 0.03$ [16,18]. $La^{3+}$, $Ce^{3+}$, $Sm^{3+}$, $Gd^{3+}$, $Dy^{3+}$ and $Yb^{3+}$ ions can be incorporated into CaHap crystals at $Ln/(Ln + Ca) \leq 0.01$, and a mixture of LnHap and $LnPO_4$ was formed at $Ln/(Ln + Ca) > 0.01$ [16,18]. It was reported that the ion exchange of $Ca^{2+}$ by $La^{3+}$ would be unlikely [19]. Cerium–calcium hydroxyapatite (CeCaHap) solid solution particles with $Ce/(Ce + Ca) \leq 0.20$ were obtained using the sol–gel method [20], and CeCaHap solid solution particles with $Ce/(Ce + Ca) \leq 0.10$ were obtained using the hydrothermal method [21]. It is generally agreed that the cationic selectivity of hydroxyapatite (Hap) depends on the radii of the cations [22]. The radii of $Ca^{2+}$ ions are 0.100 nm [23]. It is considered that cations with ion radii close to those of $Ca^{2+}$ ions are easily incorporated into CaHap crystals.

Among various MCaHaps, cerium–calcium hydroxyapatite (CeCaHap) and TiCaHap have ultraviolet (UV) absorptive abilities [17,18,24,25]. The UV spectrum is classified as UVC (100–280 nm), UVB (280–320 nm) and UVA (320–400 nm). Exposure to UV rays adversely affects humans [26–28]. UVC rays are almost completely absorbed by the ozone layer in the upper part of the Earth's atmosphere, although UVC rays cause DNA damage. UVB may cause skin cancers and sunburn. Although the influence of UVA on humans is relatively mild, UVA will cause sun tan and wrinkles on our skin. Therefore, it is necessary for us to be protected from UVB and UVA rays. $Ce^{3+}$ and $Ti^{4+}$ absorb UVA and UVB, respectively. CeCaHap particles and TiCaHap particles also absorb UVA and UVB, respectively [17,18,24,25]. However, cerium is a rare earth metal and is used for many purposes, including in abrasives, phosphors, catalysts, coloring agents, and fluorescent substances.

$Zn^{2+}$-containing Haps also have a UVA absorptive ability [26]. $Zn^{2+}$-containing Hap was prepared via the sol–gel method [29–31], $Zn^{2+}$ ion-doping method [26,32,33] and precipitation method [34,35]. Biological applications have been studied. Antimicrobial activities against *C. albicans* and *S. aureus* were investigated using $Zn^{2+}$-containing Hap $(Zn/(Zn + Ca) = 0.01)$ [29]. Antibacterial activities against *K. pneumoniae*, *B. cereus* and *S. aureus* were investigated using Zn- and Cu-doped Hap, for which an XRD pattern of 10% Zn/Cu-doped Hap showed a different patten from that of pure Hap [30]. The biocompatibility of $Zn^{2+}$-, $Mn^{2+}$- and $Fe^{2+}$-substituted Hap was investigated in which the XRD pattern of 5% Zn-Hap showed a pattern of a mixture of CaHap and another crystal phase [31]. The incorporation of bovine serum albumin (BSA) protein with Zn- and Mg-doped Hap was investigated in which the $Zn/(Zn + Ca)$ ratio of the particles was 0.005 [32]. The antifungal activity of a Zn-doped Hap thin layer was investigated [33]. ZnO-CaHap composites were prepared, and UV light illumination resulting from Zn was investigated, but zinc–calcium hydroxyapatite (ZnCaHap) solid solution particles were never prepared [34]. The antimicrobial activity of zinc-modified Hap was investigated, and the $Zn/(Zn + Ca)$ ratios of the particles were 0.01, 0.05 and 0.09 [35]. $Zn^{2+}$-, $Fe^{3+}$- and $Cr^{3+}$-doped Hap were prepared, and the UV absorbance of the particles was compared, for which the Zn and Ca contents were not apparent [26].

As stated above, various $Zn^{2+}$-containing CaHap particles were prepared using different methods and their various properties were investigated. However, the limit of $Zn^{2+}$ content in ZnCaHap solid solution is not studied. Moreover, the relationship between $Zn^{2+}$ content in ZnCaHap and UV absorptive ability is not studied. The ionic radius of $Zn^{2+}$ ions is 0.074 nm [23]; it is interesting to compare with $Mg^{2+}$ ions, because the ionic radius of $Mg^{2+}$ ions is 0.072 nm close to that of $Zn^{2+}$ ions. Therefore, in this study, the particles are prepared from starting solutions with different $Zn/(Zn + Ca)$ ($[X_{Zn}]$) ranging from 0 to 1, and the structure of the obtained particles is investigated using various means. The

extent of $[X_{Zn}]$ where pure ZnCaHap can be formed is clarified. The UV absorptive ability of $Zn^{2+}$-containing CaHap particles is also shown.

We show that ZnCaHap particles have a UV absorptive ability in the present study. ZnCaHap will become good UV-shielding materials. ZnCaHap can be supported on textile, because CaHap particles are suitable for support on the textile [17,24]. UV-shielding properties of the textile will be enhanced via the treatment. Moreover, there are many other metals which have valuable properties, including antibacterial properties, magnetism and luminescence. Various MCaHap solid solution particles can become new high-performance materials in the future.

## 2. Experimental Section

### 2.1. Materials

The particles with $[X_{Zn}]$ = 0 (named as Ca100) were synthesized using a wet method according to a previous study [5,16–18], as follows. $Ca(OH)_2$ (32 mmol) was dissolved by stirring for 1 h in 1.6 L deionized water free from $CO_2$ under an $N_2$ atmosphere. $CO_3{}^{2-}$ ions can be incorporated into $OH^-$ sites to form A-type carbonate apatite, and into $PO_4{}^{3-}$ sites to form B-type carbonate apatite. A diluted $H_3PO_4$ solution (19.2 mmol) was prepared because 85% $H_3PO_4$ solution has high viscosity. The Ca/P atomic ratio was adjusted to the stoichiometric ratio of 1.67 of CaHap (Ca/P = 5/3). After the $H_3PO_4$ solution was added, precipitates were formed. The suspension was stirred at room temperature for 1 h and then aged in a 2 L screw-capped polypropylene vessel at 100 °C using an air oven (WFO-420, EYELA, Tokyo, Japan) for 2 d. The formed precipitates were filtered off, washed with 1 L deionized water and finally dried in the air oven at 70 °C for 16 h.

The $Zn^{2+}$-containing particles were prepared by adding $Zn(NO_3)_2$•$6H_2O$ in place of a portion of $Ca(OH)_2$. The total number of moles of Zn and Ca was kept at 32 mmol, and $[X_{Zn}]$ atomic ratios were 0.01, 0.05, 0.10, 0.20, 0.25, 0.30, 0.40, 0.50, 0.70 or 1. The solution pH was brought to 9.50 by the addition of 15 mol/L $NH_4OH$ when the solution pH after adding a $H_3PO_4$ solution was less than 9.50. The remainder of the procedure was the same as that for the preparation of Ca100 described above.

All chemicals mentioned above were reagent grade and were supplied by FUJIFILM Wako Pure Chemical Co., Ltd., Osaka, Japan and were used without further purification.

### 2.2. Characterization

#### 2.2.1. XRD Measurement

The crystal structure of the products was determined via powder X-ray diffraction (XRD) using a Rigaku diffractometer (SmartLab, Rigaku, Japan) with Ni-filtered CuK$\alpha$ radiation (45 kV, 200 mA). The products were ground enough and were filled in a glass cell. Start and end angles were 10° and 80°, respectively. Step was 0.004°. In order to verify incorporation of $Zn^{2+}$ ions into Hap crystals, lattice parameters *a* and *c* were obtained from the XRD peak at 2θ = 29.0° due to the (hkl) = (210) plane and 2θ = 25.9° due to the (hkl) = (002) plane, respectively, as follows:

$$a \text{ (Å)} = \sqrt{(h^2 + hk + K^2)/3} \times 1/\sin\theta$$

$$c \text{ (Å)} = \lambda l/(2 \times \sin\theta) \quad \lambda = 1.5418 \text{ (value of the apparatus used)}$$

where lattice parameters *a* and *c* are lowered by the exchange of larger cations with smaller cations in Hap crystals. The crystallite sizes of the particles were estimated using the Scherrer equation [36] and the half-height width (b) of the XRD peaks at 2θ = 29.0° due to the (210) plane, and 2θ = 25.9° due to the (002) plane, as follows:

$$L \text{ (Å)} = \lambda/(\beta \cos\theta)$$

where K = 0.9, β = b − B, B = 0.16, λ = 1.5418 (B and λ are the values of the apparatus used).

The crystallite sizes of solid particles in solution tend to lower through the incorporation of plural cations into the Hap crystals [5,9,12].

### 2.2.2. FE-SEM and EDS Analysis

The particle morphology was observed using a field emission-scanning electron microscope (FE-SEM, JSM-7000F, JEOL, Tokyo, Japan) at an accelerating voltage of 10 kV. A small amount of the samples was dispersed in deionized water, centrifuged for 1 min and settled for 5 min. Then, the supernatant was picked up and put on a thin glass plate using a pipette. The atomic contents of the particles of $Ca^{2+}$ and $Zn^{2+}$ were obtained using an energy dispersive X-ray spectrometer (EDS, JED-2300, JEOL, Tokyo, Japan) attached to the FE-SEM at an accelerating voltage of 20 kV. The samples were dried at 100 °C for 7 d before the observation and the analysis to prevent electrostatic charge during the measurements.

### 2.2.3. UV—Vis Spectroscopy

Diffuse reflection UV—Vis spectra of the particles were obtained using a UV—Vis spectrophotometer (UV—Vis, V-670ST, Jasco, Tokyo, Japan) between 260 and 800 nm with an integrating sphere (ISV-722). The particles were ground, because the UV–Vis spectrum of the sample which was not sufficiently ground showed higher reflectance. Cells for powders with a diameter of 1.6 cm were filled with 0.3 g of each sample. The measurement conditions were as follows: band width was 2.0 nm, response was 0.24 s, data acquisition interval was 1.0 nm, scanning mode was continuous mode and scanning speed was 400 nm/min.

## 3. Results and Discussion

### 3.1. Crystal Structure and Lattice Parameter

XRD measurements were performed to determine the crystal phases of the products. Figure 1 shows the patterns of the particles formed at different $[X_{Zn}]$ from 0 to 1. Ca100 particles were identified as CaHap (JCPDS 9-432). No peaks other than Hap were found for the particles formed with $[X_{Zn}]$ = 0.01–0.20. However, some small peaks appeared for the particles formed at $[X_{Zn}]$ = 0.25, as shown by triangles in Figure 1. The peak at $2\theta$ = 10.8° of the particles with $[X_{Zn}]$ = 0–0.20 was assigned to the (100) plane of calcium hydroxyapatite. The peak at $2\theta$ = 10.3° of the particles with $[X_{Zn}] \geq 0.25$ was considered to be another phase due to a Zn compound. The two peaks might have been different crystal phases, although the positions of the two peaks were close with each other. The peaks other than Hap increased and grew with increasing $[X_{Zn}]$ from 0.25 to 0.70. The products formed at $[X_{Zn}]$ = 0.25–0.70 were considered to be a mixture of Hap and other crystals. The pattern of the products formed at $[X_{Zn}]$ = 1 was not Hap (shown by squares). Therefore, pure ZnCaHap solid solution particles were formed with $[X_{Zn}] \leq 0.20$. CaHap is hexagonal with space group $P6_3/m$ and the cations occupy two different sites, cation(I) and cation(II) sites. The cation(I) sites are columnar sites and are coordinated by nine O atoms situated in six different $PO_4^{3-}$ tetrahedra. The cation(II) sites are triangular sites and have an irregular seven-fold coordination with six O atoms of five $PO_4^{3-}$ ions and an $OH^-$ ions [22]. It has been reported that the larger cations occupy preferentially the cation(II) sites and the smaller cations prefer the cation(I) sites in MCaHap solid solution [12,22]. The incorporation of $Zn^{2+}$ ions, which are rather smaller than $Ca^{2+}$ ions, into the cation(I) sites may cause disorder of the column, leading to a destruction of the Hap structure. It can be said that the limit of $Zn^{2+}$ content to maintain Hap structure is a Zn/(Zn + Ca) ratio of 0.2 in the present study. The crystal phases other than Hap shown by triangles and squares in Figure 1 cannot be identified at the moment.

The size limit for substitution in fluorapatite ($M_{10}(ZO_4)_6F_2$) and chlorapatite ($M_{10}(ZO_4)_6Cl_2$) was reported on the basis of the ionic radii of cations [37]. It can be considered that MCaHap solid solutions are easily formed, when cations with ion radii close to that of $Ca^{2+}$ ions are used. However, ZnCaHap solid solution can be formed at Zn/(Zn + Ca) $\leq$ 0.2 in the present study and MgCaHap solid solution can be formed at Mg/(Mg + Ca) $\leq$ 0.50 in the previous study [12]. Ionic radii of $Ca^{2+}$, $Zn^{2+}$ and $Mg^{2+}$ ions are 0.100 nm, 0.074 nm and

0.72 nm, respectively [23]. Therefore, the size limit of cations which are formed by MCaHap solid solution cannot be explained only by the ionic radii of cations.

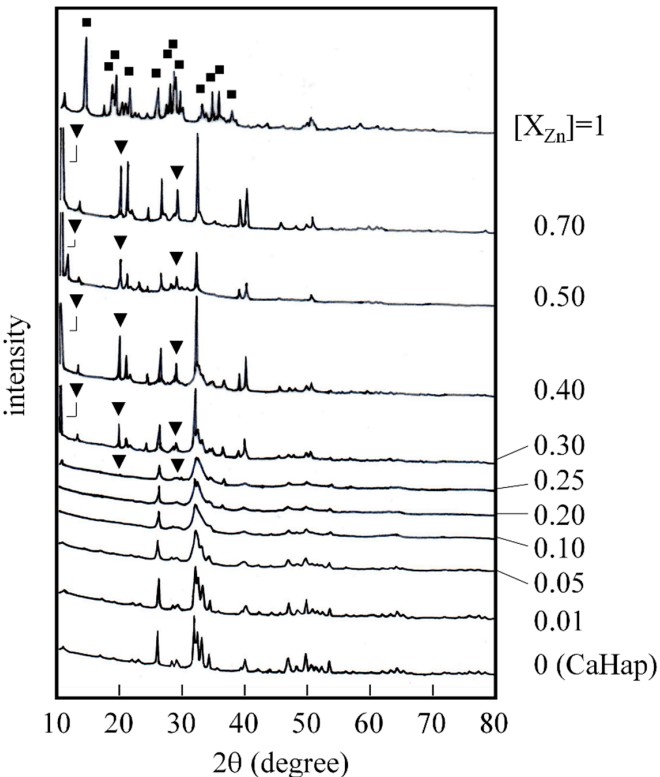

**Figure 1.** XRD patterns of various particles. ▼, ■: unknown material.

Lattice parameters *a* and *c* obtained from the XRD peaks due to the (210) and (002) planes, respectively, are shown against $[X_{Zn}]$ in Figure 2. The parameter *a* of the particles formed at $[X_{Zn}] > 0.10$ and the parameter *c* of the particles formed at $[X_{Zn}] > 0.20$ cannot be obtained because of the broadness of the XRD peaks. The parameters *a* and *c* of Ca100 with $[X_{Zn}] = 0$ were 0.9409 nm and 0.6874 nm, respectively. Both parameters decreased with increasing $[X_{Zn}]$ from 0 to 0.10. This was because larger $Ca^{2+}$ ions (0.100 nm) were replaced by smaller $Zn^{2+}$ (0.074 nm) ions [23]. It was confirmed that $Zn^{2+}$ ions are able to enter into Hap crystal and form ZnCaHap solid solution from the result of the change of lattice parameters shown in Figure 2.

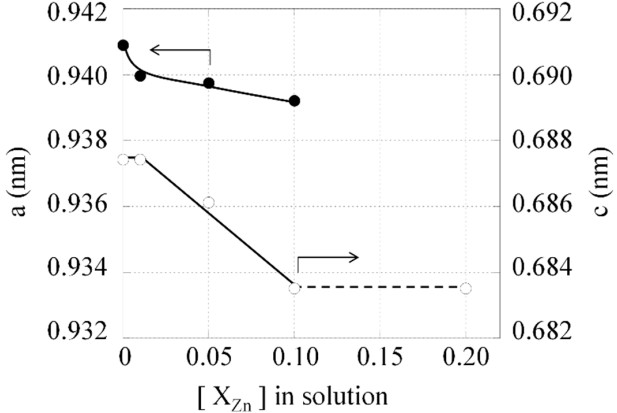

**Figure 2.** Lattice parameters *a* (●) and *c* (○) vs. $[X_{Zn}]$. Solid line: Confirmed through experiment and estimation of lattice parameters. Dashed line: Appearance from XRD patterns because of broadness of the patterns. The arrows express each vertical axis.

### 3.2. Morphology

Figure 3 presents FE-SEM images of the particles obtained at different $[X_{Zn}]$. Ca100 particles with $[X_{Zn}] = 0$ were fine ellipsoidal particles. No apparent changes were shown for the particles obtained with $[X_{Zn}] = 0.01$–0.25 compared with Ca100. The long rod-like particles were mixed with the fine particles at $[X_{Zn}] = 0.30$–0.40, although they cannot be seen in the images with high magnification. The length of the long rod-like particles reached 40 μm for the particles with $[X_{Zn}] = 0.30$, as shown by the enlarged image in Figure 3. The large plate-like particles were mixed with the fine particles at $[X_{Zn}] = 0.50$–0.70. Only large plate-like particles were formed under the conditions of $[X_{Zn}] = 1$. The morphology of the large plate-like particles with $[X_{Zn}] = 0.50$–1 was almost square, and the length of one side of the square was 0.5–2.0 μm. The size of the fine ellipsoidal particles will be mentioned in the next section. It is considered that the large plate-like particles are zinc compounds and are shown as unknown crystals in XRD patterns in Figure 1.

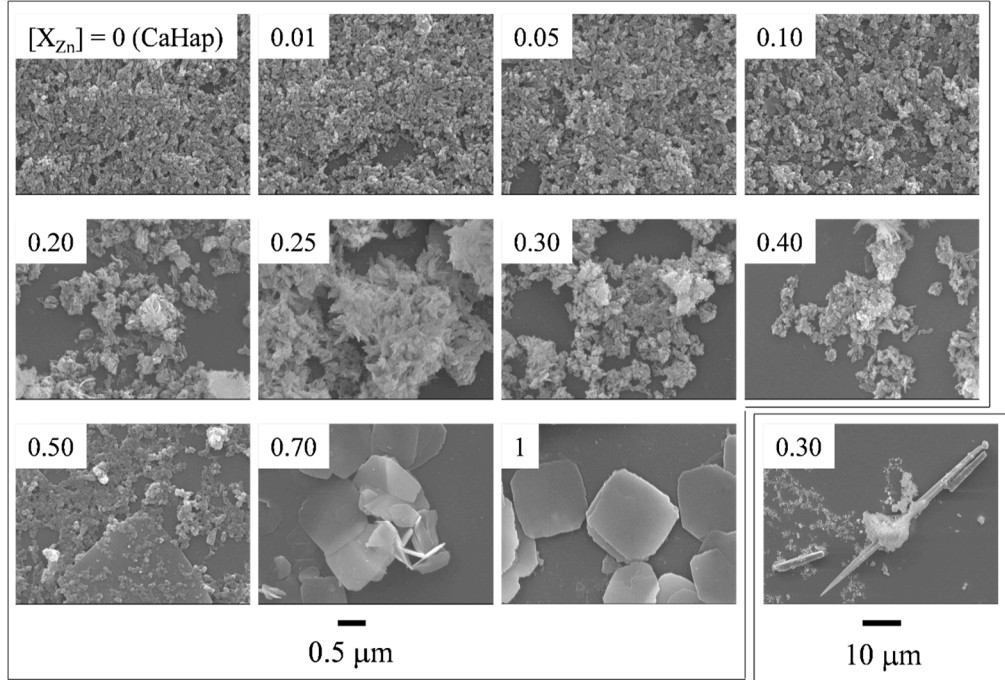

**Figure 3.** FE-SEM images of various particles.

### 3.3. Particle Sizes and Crystallite Sizes

The particle length and width of the fine ellipsoidal particles were obtained from FE-SEM images, and the results are shown in Figure 4 via open symbols as a function of $[X_{Zn}]$. Ca100 particles with $[X_{Zn}] = 0$ were $77.3 \pm 27.8$ nm in length (open circles) and $37.5 \pm 8.0$ nm in width (open triangles). The particle length decreased once at $[X_{Zn}] = 0$–0.10 and then increased at $[X_{Zn}] = 0.10$–0.20. In contrast, the particle width did not change drastically for all the particles at $[X_{Zn}] = 0$–0.20. The incorporation of $Zn^{2+}$ ions into the ZnCaHap particles influenced the particle length more than the width. The same result was observed for TiCa-Hap [17] and various rare earth metals such as calcium hydroxyapatite (LnCaHap) [16,18]. $Zn^{2+}$ ions may have been adsorbed to the side plane (a- and/or b-axis directions) of the ellipsoidal ZnCaHap particles, and crystal growth along the (a00) plane was inhibited as the crystal grew along the (00c) plane. It was reported that similar changes in Hap particles along the c-axis were caused by the incorporation of $Ce^{3+}$, $Cr^{3+}$ and $Fe^{3+}$ ions [20,21,38]. In the present study, $Zn^{2+}$ ions may adsorb to the side plane (a- and/or b-axis directions) of the ellipsoidal-shaped ZnCaHap particles as the crystal growth along the (a00) plane was inhibited and the growth along the (00c) plane was influenced.

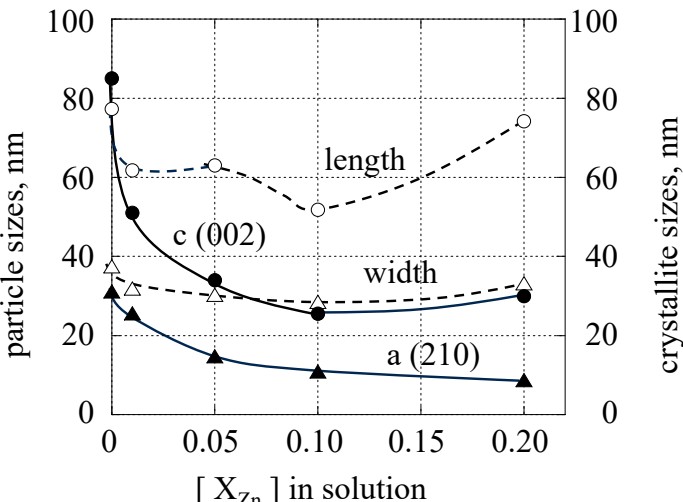

**Figure 4.** Plots of mean particle sizes ($\bigcirc$, $\Delta$ and dashed lines) and crystallite sizes ($\bullet$, $\blacktriangle$ and solid lines) of obtained particles versus [$X_{Zn}$]. $\bigcirc$: particle length, $\Delta$: particle width, $\bullet$: crystallite size due to (002) plane and $\blacktriangle$: crystallite size due to (210) plane.

The crystallite sizes of the particles were estimated from the XRD peaks due to the (002) and (210) planes. In Figure 3, the results are shown by closed circles and triangles, respectively. Both crystallite sizes decreased with increasing [$X_{Zn}$]. The size obtained from the (002) plane was more significantly changed than the size obtained from the (210) plane. From a comparison between the particle sizes and the crystallite sizes, it can be said that Ca100 with [$X_{Zn}$] = 0 is a single crystal and that other particles with [$X_{Zn}$] = 0.01–0.20 are polycrystalline particles.

### 3.4. $Zn^{2+}$ and $Ca^{2+}$ Content

The $Zn^{2+}$ and $Ca^{2+}$ contents of the particles were obtained. The Zn/(Zn + Ca) and Ca/(Zn + Ca) atomic ratios in the particles are plotted against [$X_{Zn}$] in the starting solution in Figure 5. The results of the fine ellipsoidal particles are shown by open symbols. The $Zn^{2+}$ (open circles) increased and the $Ca^{2+}$ content (open triangles) decreased with an increase in [$X_{Zn}$] = 0–0.50. This result confirmed that $Zn^{2+}$ ions added into the synthesis solution were incorporated into the particles.

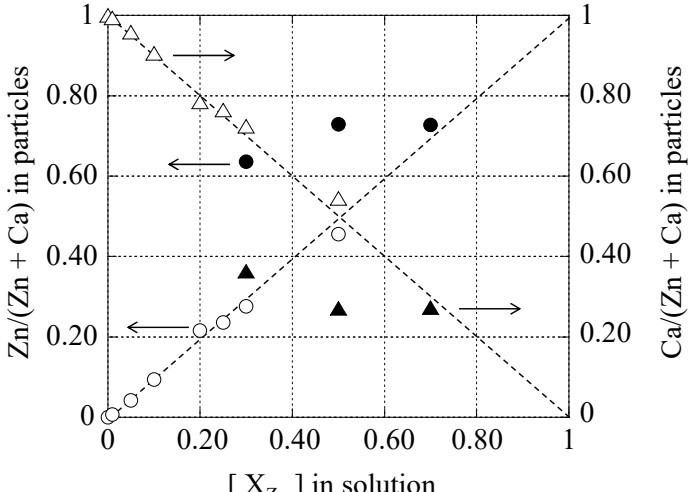

**Figure 5.** Zn and Ca contents in particles vs. [$X_{Zn}$] in starting solution. Zn: $\bigcirc$, $\bullet$ and. Ca: $\Delta$, $\blacktriangle$. Fine particles: $\bigcirc$, $\Delta$ and long rod-like and plate-shaped particles: $\bullet$, $\blacktriangle$. The arrows express each vertical axis.

As shown in the FE-SEM images (Figure 3), the products with $[X_{Zn}]$ = 0.30–0.70 are mixtures of fine ellipsoidal particles and long rod-like or large plate-like particles. Therefore, EDS analysis was separately performed for fine particles and other particles. The $Zn^{2+}$ and $Ca^{2+}$ contents of the long rod-like or large plate-like particles are shown by closed symbols in Figure 5. The $Zn^{2+}$ contents (closed circles) were higher than the $Ca^{2+}$ contents (closed triangles) for the particles with $[X_{Zn}]$ = 0.30, 0.50 and 0.70. No large changes in the contents of $Zn^{2+}$ or $Ca^{2+}$ were shown among the particles with $[X_{Zn}]$ = 0.30, 0.50 and 0.70. Therefore, $Zn^{2+}$ ions mainly exist in ZnCaHap, and the long rod-like and large plate-like particles are compounds containing abundant $Zn^{2+}$. These results do not contradict the discussion in Section 3.2.

### 3.5. UV Absorptive Ability

The diffuse reflection UV—Vis spectra of the particles prepared with various $[X_{Zn}]$ are shown in Figure 6. Ca100 particles with $[X_{Zn}]$ = 0 never exhibit strong UV absorption. As $Zn^{2+}$ ions were incorporated into the particles, the reflectance decreased remarkably in the UV range at approximately 360 nm for the particles obtained at $[X_{Zn}]$ = 0–0.30. However, no apparent changes were detected between the UV spectrum of the particles with $[X_{Zn}]$ of 0.30 and the spectrum with $[X_{Zn}]$ of 0.40. The particles formed at $[X_{Zn}]$ > 0.40 do not have UV absorptive ability. It may be that the particles with $[X_{Zn}]$ = 0.3 and 0.4 are mixtures of ZnCaHap which contains more $Zn^{2+}$ ions than ZnCaHap with $[X_{Zn}]$ = 0.2 and crystal phase other than Hap. High UVA absorbing abilities of the particles formed with $[X_{Zn}]$ = 0.3 and 0.4 may result from ZnCaHap containing more $Zn^{2+}$ ions than ZnCaHap with $[X_{Zn}]$ = 0.2.

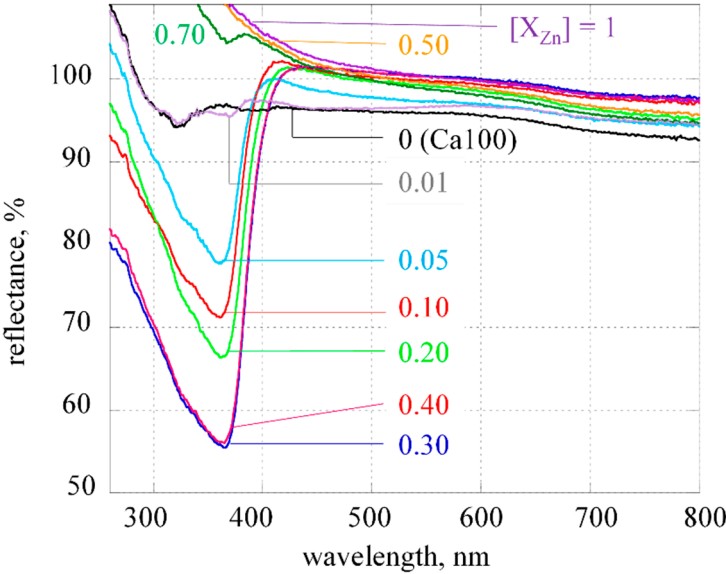

**Figure 6.** UV–Vis spectra of various samples (reflection method).

UV absorptive ability of various TiCaHap particles was measured [17]. The UV absorption increased with an increase in $Ti^{4+}$ content at Ti/(Ti + Ca) $\leq$ 0.40 and no apparent changes were shown at Ti/(Ti + Ca) > 0.40. The reason that UV absorptive ability disappears in the present study with an increase in $Zn^{2+}$ content at Zn/(Zn/Ca) > 0.50 is unclear at the moment. It can be said that fine ellipsoidal particles are ZnCaHap having UV absorptive ability, and long rod-like and large plate-like particles are zinc compounds without UV absorptive ability, from the results of XRD, FE-SEM and UV–Vis spectra.

We reported that CeCaHap has a UVA absorptive ability and that TiCaHap has a UVB absorptive ability in the previous studies [17,18,24]. ZnCaHap obtained in the present study has the UV absorptive ability of UVA, similar to CeCaHap. The UVA absorptive ability of ZnCaHap may be slightly weaker than that of CeCaHap. However, Ce is one of the rare earth metals and has various commercial uses, including abrasives, phosphor,

catalyst, coloring agent and fluorescent substance. Therefore, it is worth using a base metal such as Zn as a UVA absorber. ZnCaHap will become a promising material in the future. ZnCaHap will be able to be supported on textile, because size, shape and other properties of CaHap particles are suitable for support on textile [17,24]. UV-shielding properties of the textile will be enhanced via the treatment. Moreover, there are many other metals which have valuable properties, including antibacterial properties, magnetism and luminescence. If various MCaHap solid solution particles are prepared, new high-performance materials can be made in the future.

## 4. Conclusions

ZnCaHap solid solution particles were prepared via a precipitation method. Pure ZnCaHap particles were obtained from the solutions with $[X_{Zn}] \leq 0.20$. ZnCaHap particles were fine ellipsoidal particles. Long rod-like particles were mixed with the fine particles at $[X_{Zn}] = 0.25$–$0.40$. Large plate-like particles were mixed with the fine particles at $[X_{Zn}] \geq 0.50$. $Zn^{2+}$ ions mainly existed in the long rod-like and large plate-like particles. The $Zn^{2+}$ ions incorporated into the ZnCaHap crystal affected the particle length more strongly than the particle width. The crystallinity of the particles was lowered at $[X_{Zn}] = 0$–$0.10$ with an increase in $Zn^{2+}$ ions in the particles. The particles began to exhibit a UVA absorptive ability by the incorporation of $Zn^{2+}$ ions, and the ability showed a maximum at $[X_{Zn}] = 0.30$.

**Author Contributions:** Conceptualization, A.Y.; methodology, A.Y.; validation, A.Y.; investigation, M.Y.; writing—original draft preparation, A.Y.; writing—review and editing, A.Y.; visualization, M.Y., A.Y.; supervision, A.Y.; project administration, A.Y.; funding acquisition, A.Y. All authors have read and agreed to the published version of the manuscript.

**Funding:** This work was supported in part by JSPS KAKENHI Grant number 22K02153.

**Data Availability Statement:** Data are contained within the article.

**Conflicts of Interest:** The authors declare no conflict of interest.

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
