# Peer review of "Preparation and Structure of Zinc–Calcium Hydroxyapatite Solid Solution Particles and Their Ultraviolet Absorptive Ability"

_colloids, doi:10.3390/colloids7040070_

Round 1
Reviewer 1 Report
Comments and Suggestions for Authors
This work is dedicated to the influence of zinc ion (Zn2+) substitution on calcium ions (Ca2+) on the growth of hydroxyapatite (CaHap) crystals from aqueous solutions, their structure, morphological quality, and optical properties regarding their ability to absorb ultraviolet radiation. Materials based on hydroxyapatite with varying Zn2+ content were obtained. The structure and morphology of the materials' particles obtained were carefully studied using powder X-ray diffraction and scanning electron microscopy. The chemical composition of the materials was analyzed using energy-dispersive X-ray spectroscopy (EDS), and the diffuse reflection UV-vis spectra of particles with different Zn2+ compositions were investigated.
In my opinion, the work is done at a high scientific level and is well illustrated with easily readable images. The relevance of the work is well justified in the introduction. The content of the article is presented at a good accessibility level. The conclusions drawn correspond to the obtained results. To complete the picture, in my opinion, further research on the photostability and thermal stability of zinc-substituted hydroxyapatite crystals, conducted by synchronous DSC and TGA analysis, is needed. Additionally, considering that platelet-shaped single crystals with a square form and [Xzn] = 1 reach sizes of 200 μm, it is highly likely that the structure of these crystals can be decrypted using Single Crystal X-ray diffraction. Nonetheless, the work has left a good impression, and I recommend it for publication in Colloids and Interfaces.
Author Response
We are thankful for your beneficial comments on our manuscript.
The manuscript was rewritten taking account of suggestions by reviewers and editors and resubmitted to “Colloids and Interfaces”.
The changes made are shown by yellow highlight in the manuscript.
Our answers for your comments are as follows:
Comment: To complete the picture, in my opinion, further research on the photostability and thermal stability of zinc-substituted hydroxyapatite crystals, conducted by synchronous DSC and TGA analysis, is needed. Additionally, considering that platelet-shaped single crystals with a square form and [XZn] = 1 reach sizes of 200 μm, it is highly likely that the structure of these crystals can be decrypted using Single Crystal X-ray diffraction.
Answer: Thank you for your advice. We also want to investigate the particles obtained in the present study using synchronous DSC and TGA analysis and Single crystal X-ray diffraction. However, we do not have thus apparatuses and enough money for test requests, unfortunately.
The extent of Zn/(Zn + Ca) ratio where pure ZnCaHap particles are formed can be clarified by various means shown in the present study. Therefore, we thank you, if you agree to the present form.
Answer

Reviewer 2 Report
Comments and Suggestions for Authors
The article entitled « Preparation and Structure of Zinc-Calcium Hydroxyapatite Solid Solution Particles and Their UV Absorptive Ability» by Yasukawa and Yamada investigates the solid-state behavior of mixture of hydroxyapatite particles upon substituting Ca2+ by Zn2+. They show the formation of a solid solution at [XZn] < 0.2 and demonstrate its interesting UVA shielding abilities. Overall, the manuscript is very well structured, well written and is easy to follow. The writing style of the authors is very rigorous and factual, this is much appreciated. The article is well suited for publication prior to minor revision.
I essentially have two questions:
(i) Above [XZn]=0.2, it seems from the SEM images that another phase crystallizes. From my reading, I am not sure to have understood if this phase exists until [XZn]=1, or if a third phase appears at [XZn]=1. The XRD seems to show that at [XZn]=1 the black squares belong to a different phase from the black triangle. How does these phases would relate to the long rods or large platelets observed afterward? Then, if a new phase appears, the EDS-SEM results seems to indicate constant chemical composition: does that mean a stoichiometric compound exists in this binary system? The authors use the word “maldistributed” twice in the article, but I am not too sure about the meaning of this word in this context.
(ii) I do not understand why samples with [XZn]=0.3 and 0.4 still exhibit significant UVA absorbing abilities because at these compositions the system should already be a mixture between solid solution particles and the other unknown phase(s). Is it possible to have an enhancement of the UVA properties due to the mixture of particles?
I have also noted minor issues:
(i) The font size in the 2nd introduction paragraph suddenly becomes larger.
(ii) In the 3rd paragraph of the introduction, the authors states that “Zn2+” has UVA properites – I think that would depend of the chemical environment and I suppose the authors meant “Zn2+ inside HAP structure”?
(iii) In the 4th paragraph of the introduction, L7, I think there is an omission of one chemical ion before the 2+ charge.
(iv) In the 2.2.3 section of the material section, L5, “1.6 cm” is followed by the Greek letter phi and I think this could be removed.
Comments on the Quality of English LanguageN/A
Author Response
We are thankful for your beneficial comments on our manuscript.
The manuscript was rewritten taking account of your suggestions and resubmitted to “Colloids and Interfaces”.
The changes made are shown by yellow highlight in the manuscript. Our answers for your comments are as follows:
I essentially have two questions:
(i) Above [XZn]=0.2, it seems from the SEM images that another phase crystallizes. From my reading, I am not sure to have understood if this phase exists until [XZn]=1, or if a third phase appears at [XZn]=1. The XRD seems to show that at [XZn]=1 the black squares belong to a different phase from the black triangle. How does these phases would relate to the long rods or large platelets observed afterward? Then, if a new phase appears, the EDS-SEM results seems to indicate constant chemical composition: does that mean a stoichiometric compound exists in this binary system? The authors use the word “maldistributed” twice in the article, but I am not too sure about the meaning of this word in this context.
Answer (i): Thank you for your comment. We use the word “maldistributed” as “unevenly distributed”. We altered “maldistributed” to “mainly existed” in the new version (3.4 section and Conclusion). The particles with [XZn] = 1 do not contain Ca. Therefore, they are considered Zn compound. We guessed that large plate-like particles which were seen in the particles with [XZn] = 0.50-1 were Zn compound and the long rod-like particles which were seen in the particles with [XZn] = 0.30-0.40 were not Zn compound. However, EDS analysis revealed that the long rod-like particles also contained more Zn than Ca. In addition, XRD pattern of the particles with [XZn] = 1 was different from those of other particles. Zn/(Zn+Ca) and Ca/(Zn+Ca) ratios in the particles are close with Zn/(Zn+Ca) and Ca/(Zn+Ca) ratios in the
starting solution in the fine particles. However, Zn/(Zn+Ca) ratios are constantly higher than Ca/(Zn+Ca) ratios in long rod-like and plate-like particles regardless Zn/(Zn+Ca) and Ca/(Zn+Ca) ratios in the starting solution. Therefore, we considered that long rod-like and plate-like particles were two types of Zn compound.
The aim of this study is to clarify the extent of Zn/(Zn + Ca) ratio where pure ZnCaHap can be formed. Therefore, we thank you, if you agree to the present form.
(ii) I do not understand why samples with [XZn]=0.3 and 0.4 still exhibit significant UVA absorbing abilities because at these compositions the system should already be a mixture between solid solution particles and the other unknown phase(s). Is it possible to have an enhancement of the UVA properties due to the mixture of particles?
Answer (ii): Thank you for your question. We consider that the particles with [XZn] = 0.3 and 0.4 are mixtures of ZnCaHap which contains more Zn2+ ions than ZnCaHap with [XZn] = 0.2 and crystalline phase other than Hap. High UVA absorbing abilities of the particles with [XZn] = 0.3 and 0.4 may result from ZnCaHap containing more Zn2+ ions than ZnCaHap with [XZn] = 0.2. We added our view in the text in the new version.
I have also noted minor issues:
(i) The font size in the 2nd introduction paragraph suddenly becomes larger.
Answer (i): Thank you for your comment. The font type and font size were changed automatically when PDF was formed. However, it was our mistake that we did not notice them. We unified the font type and font size in the new version.
(ii) In the 3rd paragraph of the introduction, the authors states that “Zn2+” has UVA properites – I think that would depend of the chemical environment and I suppose the authors meant “Zn2+ inside HAP structure”?
Answer (ii): Thank you for your useful advice. It is very possible. We altered “Zn2+ also have a UVA absorptive ability” to “Zn2+-containing Hap also have a UVA absorptive ability” in the new version.
(iii) In the 4th paragraph of the introduction, L7, I think there is an omission of one chemical ion before the 2+ charge.
Answer (iii): Thank you for your indication. We altered “2+” to “Fe2+” in the new version. We are very sorry for our rudimentary mistake.
(iv) In the 2.2.3 section of the material section, L5, “1.6 cm” is followed by the Greek letter phi and I think this could be removed.
Answer (iv): Thank you for your advice. The expression of the diameter repeated in the old version. We deleted “phi” in the new version.

Reviewer 3 Report
Comments and Suggestions for Authors
The manuscript titled “Preparation and Structure of Zinc-Calcium Hydroxyapatite Solid Solution Particles and Their UV Absorptive Ability” prepared particles with different morphologies of ZnCaHap using a precipitation method and studied their crystal structure. The UV absorption ability of the particles was also measured. The manuscript can be accepted for publication after addressing following remarks:
1. Some details need to be modified:
a) the author needs to modify the formatting issues in the manuscript. (For example, Section 1, paragraph 2: the font size used in this paragraph is inconsistent with other paragraphs.)
b) there are redundant punctuations in the manuscript. (Section 2.2.3, “sec.,”, footnote for Figure 4, etc.)
2. The images in the manuscript can be made better:
a) the images in the manuscript are not clear, and the open circles and open squares in the images are blurry and difficult to read clearly.
b) The numbers in Figure 3 look like they have been stretched. I am not sure if Figure 3 as a whole (including the image) has undergone the same stretching treatment. If the image is stretched, the size of the sample in the image may be inaccurate.
3. The author should provide complete information on the reagents and samples used in the experiment (The last paragraph of Section 2.1), which can be presented in a table format.
4. Some peaks in Figure 1 are not obvious. For example, when 2θ=10°, the peaks of [XZn]=0.20 and [XZn]=0.10 seem to disappear here. The same phenomenon also occurs near 2θ=10° ([XZn]=0.10 and [XZn]=0.05). The author should explain this phenomenon appropriately.
5. For the representation of [XZn], the number of digits should be consistent. The [XZn] in the images retains two decimal places, while the number of digits in the manuscript often changes.
Comments on the Quality of English LanguageThe quality of English language is good.
Author Response
We are thankful for your beneficial comments on our manuscript.
The manuscript was rewritten taking account of your suggestions and resubmitted to “Colloids and Interfaces”.
The changes made are shown by yellow highlight in the manuscript. Our answers for your comments are as follows:
1. Some details need to be modified: a) the author needs to modify the formatting issues in the manuscript. (For example, Section 1, paragraph 2: the font size used in this paragraph is inconsistent with other paragraphs.)
Answer 1 a): Thank you for your comment. The font type and font size were changed automatically when PDF was formed. However, it was our mistake that we did not notice them. We unified the font type and font size in the new version. b) there are redundant punctuations in the manuscript. (Section 2.2.3, “sec.,”, footnote for Figure 4, etc.) Answer 1 b): Thank you for your suggestion. We altered “sec.” to “second” in section 2.2.3 and deleted “.” after “sizes” in caption of Figure 4. The redundant periods, semicolons and so on in references’ list were deleted. We are very sorry for our mistakes.
2. The images in the manuscript can be made better: a) the images in the manuscript are not clear, and the open circles and open squares in the images are blurry and difficult to read clearly.
Answer 2 a): Thank you for your suggestion. We altered open and closed squares to open and closed triangles in Figure 4 and Figure 5. In addition, we increased the size of all the symbols in Figure 4 in the new version. b) The numbers in Figure 3 look like they have been stretched. I am not sure if Figure 3 as a whole (including the image) has undergone the same stretching treatment. If the image is stretched, the size of the sample in the image may be inaccurate. Answer 2 b): Thank you for your useful suggestion. The ratio of the length and breadth of Figure 3 was changed when the image was inserted in Word file. We inserted the image in Word file again and confirmed the ratio of the length and breadth by the comparison with the original image in the new version.
3. The author should provide complete information on the reagents and samples used in the experiment (The last paragraph of Section 2.1), which can be presented in a table format.
Answer 3: Thank you for your comment. The expression “All chemicals” in the old version might be vague. We wrote detailed information of all chemicals several lines above in the text in the old version. We added “mentioned above” after “All chemicals” in the new version. We want to inform the chemicals used in the present study without a table format. We are thankful, if you accept it.
4. Some peaks in Figure 1 are not obvious. For example, when 2θ=10°, the peaks of [XZn]=0.20 and [XZn]=0.10 seem to disappear here. The same phenomenon also occurs near 2θ=10° ([XZn]=0.10 and [XZn]=0.05). The author should explain this phenomenon appropriately.
Answer 4: Thank you for your comment. We consider that the peak at 2θ=10.8° of the particles with [XZn]=0-0.20 is assigned to (100) plane of calcium hydroxyapatite. The peak at 2θ=10.3° of the particles with [XZn]≥0.25 is another phase. As the positions of two peaks are close, it is incomprehensible, but we consider that they are different crystal phases. We added our discussion of these XRD peaks in the text in the new version.
5. For the representation of [XZn], the number of digits should be consistent. The [XZn] in the images retains two decimal places, while the number of digits in the manuscript often changes.
Answer 5: Thank you for your suggestion. The unification of the decimal point is important. We unified the decimal point of various [XZn] ratios as second place (except for 0 and 1) in the new version. We are very sorry for our rudimentary mistake.
